# Synthetic Olfactory Agonist Use in the Farrowing House to Reduce Sow Distress and Improve Piglet Survival

**DOI:** 10.3390/ani11092613

**Published:** 2021-09-06

**Authors:** Robyn Terry, Tanya L. Nowland, William H. E. J. van Wettere, Kate J. Plush

**Affiliations:** 1Department of Primary Industries and Regions, South Australian Research and Development Institute, Roseworthy Campus, Roseworthy, SA 5371, Australia; tanya.nowland@sa.gov.au; 2Davies Livestock Research Centre, School of Animal and Veterinary Sciences, The University of Adelaide, Roseworthy, SA 5371, Australia; William.vanwettere@adelaide.edu.au; 3SunPork Group, 1/6 Eagleview Place, Eagle Farm, QLD 4009, Australia; kate.plush@sunporkfarms.com.au

**Keywords:** synthetic olfactory agonist, sow welfare, farrowing, piglet survival

## Abstract

**Simple Summary:**

Within intensive livestock industries, it is important to pursue the continual improvement of animal welfare. The farrowing crate design in most piggeries is currently the best way to optimize piglet welfare and reduce piglet mortality caused by being crushed by the sow. However, some studies have shown that restriction in a farrowing crate exacerbates sow distress at specific times during farrowing at lactation, which may compromise piglet survival. Therefore, there is a need to investigate strategies that may alleviate the stress experienced by the sow due to being confined. A synthetic olfactory agonist, a commercial product that mimics naturally occurring pheromones, previously reduced aggressive interactions in sows. Therefore, this product was investigated to determine its effectiveness at reducing sow stress and improving piglet survival during birth and lactation. We found no effect on the sow’s stress hormone levels between treatments in response to a stressor. However, first-litter sows experienced a decreased farrowing duration, but this did not extend to improvements in piglet pre-weaning survival. This synthetic olfactory agonist does not appear to be a suitable tool to reduce sow distress around birth or improve piglet pre-weaning survival.

**Abstract:**

The aim of the study was to investigate if the application of a synthetic olfactory agonist (SOA) would reduce indicators of stress in sows, in response to a stressor prior to parturition, and if it would improve farrowing house performance of sows and their piglets. Two studies were conducted: an intensive study with 47 sows, either having their first or second litter (Control *n* = 24; SOA *n* = 23); and a commercial validation study with 418 sows, either having their first litter or have had multiple litters (Control *n* = 210; SOA *n* = 208). Within the intensive study, sows were housed with or without a synthetic olfactory agonist suspended in the creep area of the farrowing crate, whereas within the commercial validation study, sows were housed with or without a synthetic olfactory agonist suspended over the adjoining creep area of two farrowing crates. Within the intensive study, despite a discernible increase in cortisol concentration in response to a stressor (snout rope test), cortisol response was not different between treatments (*p* > 0.05). Farrowing duration in first-litter sows exposed to the SOA was decreased (*p* < 0.001) whilst there was no impact on farrowing duration in second litter sows. Piglets were not attracted by the SOA to increase their utilisation of the creep area and spent more time in proximity to the sow (*p* < 0.05). Within the commercial validation study, no impacts were seen on piglet production measures (*p* > 0.05). Largely the use of an SOA within the farrowing house did not impact the sow or her piglets in either the intensive study or commercial validation study. Based on these current results, the use of SOA within the farrowing house is not supported.

## 1. Introduction

Sows are housed in farrowing crates to restrict movement to reduce piglet mortality, as the primary cause of piglet death within the first few days of life is attributed to crushing by the sow [1,2]. However, the farrowing crate design restricts natural nesting behaviours and as such, has the potential to cause physiological stress [3,4,5]. Restriction of sow movement prior to farrowing may also prolong farrowing duration, leading to an increase in stillbirths [6,7,8]. Many factors can contribute to prolonged parturition in sows; however, Langendijk et al. [8] found sows with prolonged parturition may already be compromised prior to the first piglet being born, pointing to the housing environment as an underlying cause. Antepartum fetal and maternal cortisol increases naturally prior to parturition to signal the commencement of parturition and contribute to fetal development, rather than indicating a stressful event [9,10,11,12]. However, across several domestic animal species, a pronounced increase in cortisol directly before and during parturition is thought to be attributable to the pain and potential stress of giving birth, particularly in occurrences of a prolonged birth [12,13,14]. Comparing sows in farrowing crates with those able to move around with nesting material, cortisol levels were higher in crated sows prior to farrowing and the duration of farrowing was also longer in crated sows [6]. Jarvis et al. [9] reported higher levels of cortisol during the pre-farrowing period in crated sows compared with sows able to move freely. Consequently, there is a need to investigate strategies to prevent the elevation of cortisol levels beyond those involved in the parturition process.

Naturally occurring pheromones found in the sebaceous glands of the mammary tissue may have a role in attracting piglets to the udder and calming them during feeding events [15]. These pheromones are absorbed by the piglet via the nasal cavity, stimulating the hypothalamus and amygdala regions of the brain [15,16]. By isolating skin secretions from sows, Pageat [17] created a synthetic product, which we termed a synthetic olfactory agonist (SOA) as it mimics these maternal sow pheromones being largely comprised of a mixture of fatty acids. A single application of this product to either the feeder or snout of weaner piglets resulted in more time feeding, less time spent exhibiting agonistic behaviours, and subsequently improved growth from day 0 to 28 post-weaning [18]. Aggressive behaviours have also been shown to be reduced in both weaned piglets [19] and group-housed sows [20] that were expose to SOA which is suggestive of a reduced response to stressful periods. However, there is a paucity of data as to the impact of SOA or other synthetic pheromones on the behaviour of peri-parturient and lactating sows and their piglets. Therefore, the aim of the study was to determine if exposure to SOA reduced some indicators of stress in sows housed in farrowing crates. Additionally, this study aimed to determine if the use of a heat mat with the addition of the SOA would be an attractant for nursing piglets to the creep area of the farrowing crate thus reducing the incidence of piglet deaths due to crushing by the sow.

## 2. Materials and Methods

This experiment is a part of a larger study investigating strategies to reduce sow stress around farrowing [21]. This experiment was conducted in two parts: firstly, an intensive study investigating the use of SOA within a research piggery and secondly, within a commercial piggery. Animal procedures were conducted in accordance with the Australian Code for the care and use of animals for scientific purposes (NHMRC 2013) with approval from the animal ethics committee of Primary Industries and Regions South Australia (Animal Ethics Number: 25/16).

### 2.1. Animals and Measurements—Intensive Piggery Study

A total of 47 Large White x Landrace sows (*n* = 13 first litter; and *n* = 34 s litter sows) and 563 piglets were housed in climate-controlled farrowing rooms at a research piggery in Roseworthy, South Australia. Sows were moved into conventional farrowing crates approximately five days before the expected farrowing date. The farrowing crates were on fully slatted plastic flooring, measured 1.8 m × 2.4 m, and contained a heated creep mat for piglets. Sows were fed 2.5 kg/d of a commercial lactation diet prior to farrowing and gradually increased 7–8 kg by day seven of lactation and through to weaning, which occurred at 18 days.

Sows were allocated to one of two treatments over two replicates at farrowing shed entry with parity evenly stratified across treatments and replicates. The two treatments applied were: Control (CON): where no synthetic olfactory agonist was supplied (*n* = 7 first litter; and *n* = 17 s litter sows) and; synthetic olfactory agonist (SOA; SecurePig^®^): an SOA diffuser block fitted approximately 0.5 m above the creep area of each farrowing crate (*n* = 6 first litter; and *n* = 17 s litter sows) from sow entry to the farrowing crate until weaning. The placement of the SOA diffuser block can be seen in Figure 1. The active components of the SOA block were based on mammary secretions of the sow and included: methyl caprate, methyl laurate, methyl miristate, methyl palmitate, methyl linoleate, methyl oleate [15]. Given the fact the SOA treatment was an air diffuser, all SOA treated sows were confined to one room in the first replicate and Control to another. There was no shared ventilation between the two rooms. In between replicates, rooms were disinfected and allowed to rest for 48 h, before the treatments were switched between rooms.

Prior to farrowing, a subset of both first and second litter sows was selected to determine if the SOA block had an effect on their ability to cope with stress (Control: *n* = 6; SOA: *n* = 7), assessed by a snout rope test. The snout rope test, adapted from Farmer et al. [22], was conducted to cause stress to the sow by restraining the sow with a rope snare and fastening the rope to the bars of the farrowing crate for a total of five minutes. Sows were introduced to the farrowing rooms and provided with the SOA for a full three days prior to the snout rope test. The snout rope test was conducted two to five days prior to the actual farrowing date. The day prior to the snout rope test, sows had an ear vein catheter inserted (vinyl, Microtube Extrusions Pty Ltd., North Rocks, NSW, Australia), to allow for continual blood sampling. Each sow was given topical anaesthesia (Xylocaine Jelly 2% Gel, AstraZeneca Pty Ltd., NSW, Australia) on both ears at least 20 min prior to ear vein catheter insertion. The sow was restrained by a nose snare and an indwelling jugular-vein cannula via an ear vein was inserted. The catheter was secured to the neck of the sow through a small case that was secured with tape, to allow for easy sampling without disturbing the sow. At the conclusion of the test, the rope snare was removed as quickly as possible. On the day the test was conducted, sows were fed approximately 2.5 kg at 7.00 a.m. Blood samples commenced at 8.30 a.m. and concluded at 12.30 p.m. Samples were taken every 15 min for a total of 120 min prior to the snout rope test, 1 min prior to the snout rope test which was conducted for five minutes, and again every 15 min for a total of 120 min post- snout rope test. Blood samples were taken via syringe through the catheter and immediately transferred into a 5 mL Heparin-Lithium-coated collection tube (Vacuette, Greiner Labortechnik, Austria). Blood samples were maintained on ice and were centrifuged at 1500× *g* for 10 min and plasma stored at −20 °C. Cortisol samples were analysed in duplicate with a commercial radioimmunoassay kit (ImmuChemTM CT cortisol kit, MP Biomedicals, Orangeburg, NY, USA) according to the manufacturer’s instructions. The average intra- and inter-assay CV was 2.2% and 23.6%, respectively.

Sows were monitored for 24 h a day for the measurement in real-time of farrowing data by an observer. The farrowing duration was determined from the time that the first piglet was born, regarded as the commencement of farrowing, until the time of the last piglet being born, regarded as the conclusion of farrowing. Inter-piglet birth interval measurements were recorded in real-time. Piglets were assigned a birth order group of 1 for the first 4 piglets born, 2 for the next 4 piglets born and 3 for the remainder of piglets born. At birth, all piglets received an individual ear tag, were weighed, rectal temperature and meconium staining score recorded, based on methods previously published [23]. Immediately following these measures, piglets were placed back into the farrowing crate at the back of the sow and time from birth to udder and birth to first suck were recorded. Piglet mortality was recorded from birth to weaning. At 24 h of age, blood samples were collected from the first three and the last three piglets born. Blood samples were taken via venepuncture using a 21 G needle and transferred immediately into a 3 mL Heparin-Lithium-coated collection tube (Vacuette, Greiner Labortechnik, Austria). Blood samples were maintained on ice and were centrifuged at 1500× *g* for 10 min and plasma stored at −20 °C. Plasma samples were analysed for IgG concentrations (mg/mL) by using radial immunodiffusion assay at the University of Adelaide’s Vet Diagnostic Lab. Piglets were fostered within treatment at approximately 24 h after birth and litter numbers were determined based on sow teat capacity. Individual piglet weights were taken immediately after birth, and on days 1, 3 and 18 days of age. Colostrum intake was estimated based on body weight gain between birth and 24 h of age [24]. Piglets were weaned on day 18 of age (17.9 ± 0.01).

Piglet behaviour was monitored using closed-circuit television (CCTV) cameras (3-megapixel fixed lens IP dome cameras, Hikvision HDTVI Cameras, China) which were mounted directly above each farrowing crate and were connected to a 16 channel NVR system. Piglet position within the farrowing crate was monitored between 9 a.m. and 3 p.m. on days 2 and 3. At 10-min intervals, the number of piglets within the creep area or next to the sow’s udder were recorded. All analysis was performed using Observer XT v11 (Noldus Information Technology, Wageningen, The Netherlands).

### 2.2. Animals and Measurements—Commercial Validation

A second experiment was conducted to evaluate the use of the SOA within a commercial piggery located in South Australia. The commercial experiment was conducted over two replicates and between December and March 2017–2018. A total of 418 mixed parity Large White x Landrace sows (*n* = 322 first-litter sows; and *n* = 96 multiparous sows (parity range: 2 to 7; 3.4 ± 0.14)) and their piglets were housed in conventional farrowing crates from 5 days before expected farrowing until weaning which occurred at 25.0 ± 0.2 days.

Sows were allocated to one of two treatments at farrowing shed entry with parity evenly distributed across treatments and blocks. The two treatments were: control: sows farrowed in absence of SOA (*n* = 210) and; SOA: sows farrowed with an SOA diffuser block suspended over the creep area of two adjoining crates (*n* = 208).

Each replicate contained four farrowing rooms with 54 crates which were naturally ventilated. Two rooms contained the Control treatment and two others the SOA treatment. Rooms were pressure washed and disinfected to ensure product residue was removed and alternated between each replicate. The SOA diffusers were placed on the same day the sow entered the farrowing crate. All piglets were fostered within treatment. Measures recorded post-parturition were the total number of piglets born, number of piglets born alive or stillborn, piglet mortality prior to fostering and through to weaning, as well as the total number of piglets weaned. A subset of litters (*n* = 100 Control and *n* = 80 SOA) were weighed after fostering and again on day 21 of lactation.

### 2.3. Statistical Analysis

Analyses were conducted in SPSS v21 (IBM, Armonk, NY, USA). All data were tested for normality using the Shapiro-Wilk test, and transformations of the data were conducted where necessary and back-transformed means are given in parenthesis. Data are expressed as mean ± SEM. Probability values stated as being *p* < 0.05 were described as significant.

Cortisol data were sqrt transformed and analysed using GLMM model with time as the repeated measure, replicate as a random effect and treatment and sow parity as fixed effects. The farrowing duration was log10 transformed and analysed using UNIANOVA with replicate, total born, sow parity and treatment and their interaction as fixed effects. Inter-piglet birth interval and time to udder were log10 transformed and analysed using GLMM with replicate and sow as random effects, total born, birth order, sow parity and treatment and their interaction as fixed effects. Piglet mortalities were analysed using GLMM with Poisson distribution, with replicate as the random effect, treatment and sow parity and their interaction as fixed effects. Piglet location within in the farrowing crate on days 2 and 3 were analysed using GLMM with sow as the subject, time as the repeated measure, replicate as the random effect, and treatment and sow parity and their interaction as fixed effects. All other piglet traits were analysed using GLMM, with replicate and sow as a random effect, and piglet sex, sow parity and treatment and their interaction as fixed effects. Litter weights were analysed using GLMM with replicate as the random effect, sow parity and treatment and their interaction as fixed effects.

## 3. Results

### 3.1. Intensive Piggery Study

The cortisol response to the snout response was unaffected by treatment (*p* = 0.612), with sows in both treatments experiencing a significant increase in cortisol in response to the snout rope test (Figure 2). The snout rope test resulted in a significant increase in cortisol in both control and SOA treatments between −1 min and 15 min post-snare (67.7 and 76.1 nmol/mL increase in cortisol, respectively, between −1 and 15 min post snare), before returning to pre rope snare levels.

The farrowing duration was reduced for first lactation sows who were exposed to the SOA treatment compared to the Control (2.1 ± 0.08 (back-transformed mean 121.1 min) and 2.6 ± 0.08 (back-transformed mean 382.8 min), respectively, *p* < 0.001). There was however no effect of the SOA treatment on the farrowing duration for second litter sows compared to Control sows (2.3 ± 0.05 (187.9 min) and 2.3 ± 0.05 (184.5 min), respectively, *p* > 0.05).

Piglet measurements from birth until weaning are presented in Table 1. Overall, the use of SOA did not affect any piglet measures (Table 1; *p* > 0.05). Piglet weights were not different at birth or day 1 of age however, at 3 days of age, piglets exposed to the SOA were 81 g heavier than their control counterparts (Table 1; *p* < 0.05) but were similar in weight at 18 days of age (Table 1; *p* > 0.05). Amongst SOA treated sows there was a trend (Table 1; *p* < 0.068) for a shorter interval of time the piglets took from birth to reaching the udder.

A greater number of Control piglets were in the creep area, primarily during the morning, for both days 2 and 3 compared to the same time in the SOA treatments (Figure 3A,C; *p* < 0.05). Conversely, there were a greater number of SOA treated piglets located at the udder, during the same period than the control piglets for both days 2 and 3 (Figure 3B,D; *p* < 0.05).

### 3.2. Commercial Validation

Treatment had no effect on total piglets born, the number of piglets born alive or stillborn, or the number of mummified fetuses (Table 2; *p* > 0.05). Second litter sows had higher total piglets born, born alive and stillborn piglets (Table 2; *p* < 0.0001).

Litter weights on days 0 and 21 of age were not different between treatments; however, for second litter sows, litter weights on day 21 were almost 8.9 kg higher than their first-litter sow counterparts (Table 3; *p* < 0.001). Average piglet weight on day 0 and day 21 of age was lower in the SOA treatment (Table 3; *p* < 0.05). Piglet mortality in the first 24 h of life (pre-fostering) and total piglet mortality was not different between treatments or between parities (Table 3; *p* < 0.05).

## 4. Discussion

To our knowledge, this is the first study to investigate whether exposure to a synthetic olfactory agonist affects the cortisol response of peri-parturient sows to a stressor, the progression of farrowing and the growth and survival of piglets to weaning. Two studies were conducted and reported: one a small, intensive study and the other a large-scale commercial validation study. Whilst there was a clear increase in cortisol concentration in response to a stress test (a snout rope test), the magnitude of the cortisol response of the sow to this stressor was unaffected by exposure to the SOA. Despite this, farrowing duration was significantly reduced by SOA but only in first parity sows with no impact evident in second lactation sows. Piglet viability and production measures were largely unaffected by treatment. Piglet use of the creep area was lower for SOA treated piglets compared to the Control piglets on both days 2 and 3; with SOA treated piglets tending to preference being near the sow’s udder on both days 2 and 3 post-partum. Despite the SOA treated piglet’s preference to be near the sow’s udder rather than the creep area, piglet mortality pre- and post-fostering was unaffected. Given these data, our hypothesis was largely disproven with SOA unsuccessful at reducing sow stress and improving piglet survival.

The snout rope test was conducted to elicit a stress response from the sow as evidenced by a rise in plasma cortisol levels. The decision in the current study to conduct a snout rope test rather than assess cortisol concentrations during the peri-parturient period was based on the natural elevation in cortisol which occurs during parturition, and the variation in this elevation between sows [10] making it unlikely that treatment effects on cortisol would be detected. We hypothesised that the snout rope test would cause a stressful event for the sow, producing a discernible cortisol increase and enable any impacts of the SOA to be determined. All sows demonstrated an immediate increase in plasma cortisol levels following the rope snare test, similar to evidence found from Farmer et al. [22].

No previous studies have investigated whether SOA reduces the cortisol response of pre-partum sows to a known stressor (snout rope test) and confinement. Plasma cortisol release prior to parturition was unaffected by the provision of SOA. Although, the application of SOA to pens of group-housed sows at mixing reduced aggressive interactions, a concomitant reduction in salivary cortisol was not observed [20]. Conversely, salivary cortisol was decreased in adult miniature sows when a social stressor was applied following two weeks of exposure to a synthetic maternal pheromone, applied as an aerosol [25]. Notwithstanding differences in the genetics of the sows used by Yonezawa et al. [25] and those used in the current study, the lack of cortisol response to a known stressor in the current study may reflect the shorter duration of exposure prior to the imposition of the stressor (3 days versus 2 weeks), differences in the method of application (slow-release diffuser versus aerosol) and, potentially, the nature of the known stressor. Unfortunately, there is a paucity of data on the length of time considered adequate for the SOA to diffuse. Despite the pig having a highly developed sense of smell, we know relatively little about the pig’s olfactory system and how pheromones affect their behaviour [26,27]. Ideally, more research is required to understand the role of olfactory organs in the pig and how different pheromone delivery mechanisms and length of exposure can influence the outcome.

Farrowing durations in excess of 240–300 min are considered to be a stressful event for the sow, putting her piglets at greater risk of perinatal mortality [6,28]. In the first study, farrowing duration of the first litter and Control sows was, on average, greater than 6 h (range: 2.45 to 15.2 h) and significantly higher compared with the average farrowing duration of 2 h (range: 1.2 to 4.2 h) observed for first litter, SOA sows. Although unaffected by the presence of SOA, farrowing duration of second litter sows are slightly higher than values previously reported in the literature (2.5 h) and considered to be normal [29]. The cause and effect observed in this study on the farrowing duration of first-litter sows receiving the SOA may however not be determined simply due to the vastly known causes for extended farrowing duration [28,29,30,31]. However, the positive relationship between increased farrowing duration and the incidence of stillborn piglets is well documented [32]. It is, therefore, surprising that incidences of stillbirths were unaffected by parity or the provision of SOA to first lactation sows in this study; however, this may reflect the small number of first lactation sows involved. Although farrowing duration was not measured in the second, commercial study, which involved a higher number of first lactation sows, there was no effect of SOA on stillbirth rate likely indicative of a lack of an effect on farrowing duration.

A significant proportion of piglet mortality occurs in the first 3 days post-farrowing [33]. The provision of a heated creep area within the farrowing pen is designed to entice the newborn piglets away from the sow’s udder when they are not sucking, in an effort to reduce piglet crushing, injury or death. However, several studies have reported that the majority of newborn piglets will seek warmth near the sow’s udder rather than the heated creep area, potentially leading to higher mortality from sow crushing [34,35,36,37]. It has, therefore, been suggested that simply placing a heat source in the creep area does not effectively entice piglets away from the sow when not nursing [34]. It was, therefore, hypothesised that the use of a heat mat with the addition of the SOA, being a synthetic analogue of pheromones and known to be an attractant for nursing piglets [15,16], would attract more piglets to the creep area and, thus, reduce mortality. In fact, the number of piglets lying next to the sow’s udder in the 10-min intervals was increased in SOA treated crates and inversely the number of piglets in the control treatment spent more time in the creep area, although this was also dependent on the time of day. Both the usage of the creep area [35] and the prevalence of piglet deaths from crushing [35] are known to differ strongly between sows. The effect of the individual sow, and potentially the odour preferences of the piglets, may also account for where the piglet’s preference was to lay during the morning on day 2 and 3 of age and, given, the small sample size used may have affected the outcomes observed. Interestingly, despite the failure of SOA to attract piglets to the creep area and an apparent increase in piglet time spent lying next to the sow, piglet mortality was not affected in either the first (intensive) or second (commercial) study.

Whilst the use of the SOA has been shown to reduce sow aggressive interactions at mixing [20], increase weight gain in weaners [18] and reduced aggression in weaned pigs [15], unfortunately, the SOA had little effect within the farrowing house on both the sow and her piglets. The use of the SOA would also increase costs of production to the pig farmer through the purchase of diffusers and increase in labour costs to install the diffusers. However, pig farmers would adopt any practices or products to improve the welfare of their pigs, if there is a demonstrated benefit to the pig [38]. In terms of the economic feasibility, a recent study has demonstrated a 100% probability that the use of applying a synthetic maternal pheromone, for the benefit of reducing pig aggression, will result in a net loss of between GBP 0.06 and GBP 0.45 per pig [39]. Due to no currently demonstratable benefit for the use of SOA within the farrowing house, and the costs associated with the purchase and installation of the SOA, it is not recommended for pig farmers to adopt the SOA for use within the farrowing house. Other measures should be explored by pig farmers looking to better the welfare of their sows and piglets within the farrowing house.

## 5. Conclusions

In conclusion, we demonstrated that the use of the synthetic olfactory agonist within the farrowing house did not achieve a demonstratable impact on the sow or her piglets. This finding is contradictory to the small pool of scientific literature investigating the impact of this product on pigs. However, first-litter sows experienced a decreased farrowing duration which may warrant further investigation with a greater number of sows. Further studies into the effect of SOA within the farrowing house should determine if an aerosolised SOA will influence measures in the sow and her piglets. Potentially the length of exposure of the SOA block diffuser was not adequate however, this study was impeded by the practicalities of batch farrowing and may therefore also not be commercially achievable. Currently, with minimal impacts from this product, and considering the cost to purchase and labour required to install, we recommend that the synthetic olfactory agonist product not be used for commercial use in the pork industry.

## Figures and Tables

**Figure 1 animals-11-02613-f001:**
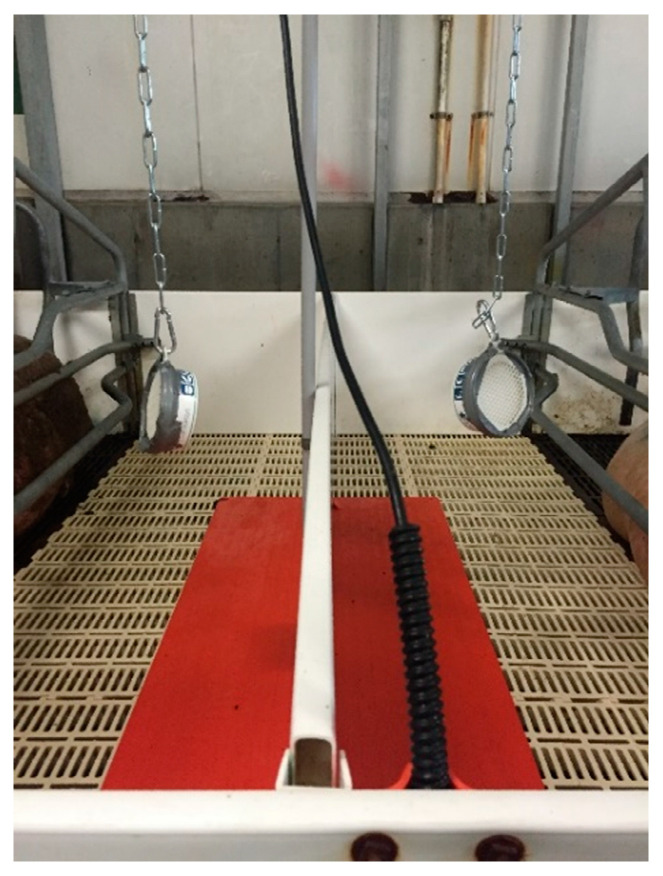
Placement of the Synthetic Olfactory Agonist diffuser blocks in the farrowing crate, above the heat mats.

**Figure 2 animals-11-02613-f002:**
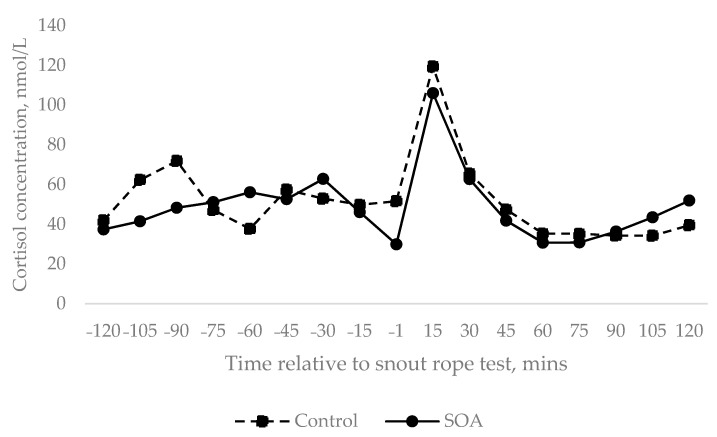
Sow cortisol concentration (nmol/L) in relation to the snout rope test which occurred at time zero. Data were sqrt transformed for analysis. Back transformed means are presented and therefore no SEM are present.

**Figure 3 animals-11-02613-f003:**
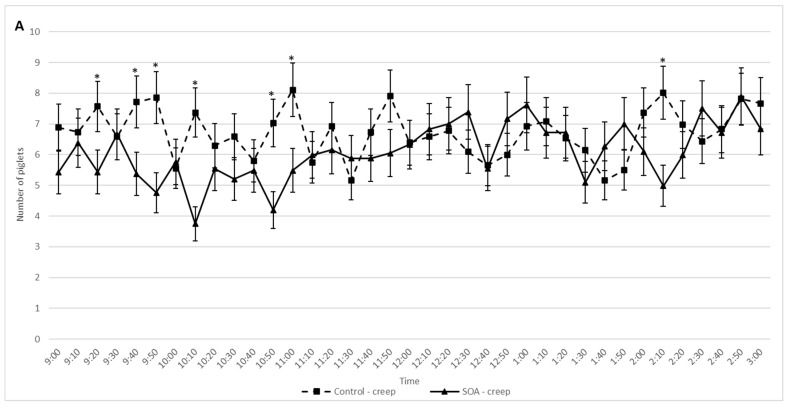
Mean ± SEM number of piglets in the creep area on day 2 of age (**A**). Mean ± SEM number of piglets next to the udder on day 2 of age (**B**). Mean ± SEM number of piglets in the creep area on day 3 of age (**C**). Mean ± SEM number of piglets next to the udder on day 3 of age (**D**). The number of piglets in either location was recorded every 10 min from video analysis from 9 a.m. to 3 p.m. * indicates significant difference between treatments at a given timepoint (*p* < 0.05).

**Table 1 animals-11-02613-t001:** Effect of the synthetic olfactory agonist (SOA) and Control treatments on piglet measurements (mean ± SEM).

	Control	SOA	*p* Value
Meconium stain score	1.3 ± 0.1	1.2 ± 0.1	0.367
Log_10_ inter piglet birth interval, min ^#^	1.1 ± 0.03 (12.11)	1.1 ± 0.04 (11.64)	0.751
Birth rectal temperature, °C	38.5 ± 0.1	38.4 ± 0.1	0.118
Log_10_ birth to udder, min ^#^	1.1 ± 0.02 (13.68)	1.1 ± 0.02 (11.99)	0.068
Birth to first suck, min	25.7 ± 1.5	22.9 ± 1.5	0.152
Colostrum intake, g	330.8 ± 7.7	346.8 ± 8.6	0.148
Plasma IgG concentration, mg/mL	68.6± 2.7	68.3 ± 2.8	0.921
Total piglets born	12.0 ± 0.5	11.5 ± 0.6	0.494
Litter size after cross fostering	11.4 ± 0.2	11.2 ± 0.2	0.543
Litter size at weaning	11.2 ± 0.2	10.8 ± 0.2	0.254
Pre-fostering mortality	0.2 ± 0.1	0.2 ± 0.1	0.838
Post-fostering mortality	0.2 ± 0.1	0.2 ± 0.1	0.964
Weight, kg			
Birth	1.4 ± 0.02	1.4 ± 0.02	0.313
Day 1	1.5 ± 0.02	1.5 ± 0.02	0.731
Day 3	1.9 ± 0.01	2.00 ± 0.01	<0.001
Day 18, weaning	6.0 ± 0.2	6.0 ± 0.2	0.758

^#^ Back-transformed means are presented in parentheses as minutes.

**Table 2 animals-11-02613-t002:** Treatment and parity effects on piglets born within the commercial validation study (mean ± SEM).

	Treatment		*p* Value	Parity		
	Control	SOA		First Litter	Second Litter	
Total piglets born	13.2 ± 0.2	13.5 ± 0.3	0.400	12.5 ± 0.2	14.3 ± 0.3	<0.0001
Piglets born alive	12.4 ± 0.2	12.4 ± 0.2	0.863	11.7 ± 0.2	13.1 ± 0.3	<0.0001
Still born piglets	0.6 ± 0.1	0.7 ± 0.1	0.301	0.5 ± 0.04	0.9 ± 0.10	<0.0001
Mummified fetuses	0.2 ± 0.03	0.3 ± 0.05	0.345	0.2 ± 0.03	0.3 ± 0.06	0.071

**Table 3 animals-11-02613-t003:** Effect of the synthetic olfactory agonist (SOA) and Control treatments on piglet measurements within a commercial research setting (mean ± SEM).

	Treatment		*p* Value	Parity		*p* Value
	Control	SOA		First Litter	Second Litter	
Litter size						
d0	11.6 ± 0.1	11.8 ± 0.1	0.093	11.8 ± 0.1	11.7 ± 0.1	0.348
d21	10.4 ± 0.2	10.4 ± 0.2	0.952	10.2 ± 0.1	10.5 ± 0.2	0.166
Weaning	10.3 ± 0.1	10.3 ± 0.1	0.728	10.2 ± 0.1	10.4 ± 0.2	0.292
Litter weight, kg						
d0	15.4 ± 0.3	14.9 ± 0.3	0.294	15.0 ± 0.3	15.2 ± 0.5	0.747
d21	59.2 ± 1.3	56.4 ± 1.4	0.124	52.9 ± 1.1	61.8 ± 1.8	<0.001
Avg piglet weight, kg						
d0	1.34 ± 0.03	1.26 ± 0.03	0.026	1.3 ± 0.02	1.3 ± 0.04	0.271
d21	5.7 ± 0.1	5.4 ± 0.1	0.024	5.2 ± 0.1	5.9 ± 0.1	<0.001
Pre-foster mortality	0.4 ± 0.1	0.4 ± 0.1	0.631	0.5 ± 0.04	0.4 ± 0.1	0.424
Total piglet mortality	1.2 ± 0.1	1.3 ± 0.1	0.678	1.4 ± 0.1	1.1 ± 0.1	0.100

## Data Availability

The data presented in this study are available on request from the corresponding author.

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
