# Peer review of "Synthetic Olfactory Agonist Use in the Farrowing House to Reduce Sow Distress and Improve Piglet Survival"

_animals, 2021, doi:10.3390/ani11092613_

Round 1
Reviewer 1 Report
Review of the manuscript No animals-1339185.
General comment
The problem of relations between sow and her piglets is one of the crucial factors determining results of rearing. It is well known that pheromones have important role in regulating swine behavior. This knowledge is commonly used is animal production. The most widespread is the use of boar pheromones to stimulate reproductive function and behavior of gilts and sows. However, there are many other areas where pheromones could be used. One of them is the influence on sow and piglets behavior. The Authors performed nice experiment to show efficiency of novel preparation containing pheromones of mammary gland. The most important advantage of this experiment is its two phase character, including experimental station and commercial conditions. The results are generally negative, however, in my opinion they are worthy to publish. The language of manuscript is clear and easy to understand. I do not feel qualified to assess language correctness (the Authors are native speakers, I do not).
Specific comments
The Introduction is well prepared and informative. I have only one remark. In my opinion the Authors should shortly describe the other areas of pheromone preparations use (especially the use of boar pheromones in sows and gilts).
The most important doubts seem to be connected to M&M, especially in comparison to data presented in Results.
Line 140. The most important parameters of RIA test should be described (sensitivity, cross reactions etc.).
Line 141. Description of the monitoring method is needed (was it camera monitoring or direct monitoring performed by observer).
Line 150-151. This sentence is inconsistent to data in table 1.
Line 192-193. Which test was used to analyze normality? Why the Authors decided to use data transformation instead of the use of non parametric tests?
Presentation of data in results also rise some doubts.
Line 223-227. The data of farrowing duration are presented after logarithmic transformation with original data in brackets, but there is no clear information about it (the reader must guess by himself). It is confusing and should be clearly described (like in table 1).
Table 1. The presentation of data should be unified (one or two points after coma should be maintained for all data).
There is lack of piglets mortality data from birth (should be completed)
There is lack of body weight in day 1 (it can be assumed that day of birth is day 0). It is important because of cross fostering litters after 24 hours and must be completed.
The data of mortality post fostering seem to be questionable. Litter size after cross fostering in control group was 11.4, and at weaning was 11.2, so the value of mortality 0.2 is OK. But in SOA group litter size after cross fostering was 11.2, and at weaning 10.8, so the difference is 0.4, not 0.2. I mortality in this group was also 0.2 (like in table 1), so the question is what was wrong with another 0.2 piglet? This must be explained.
Table 2 and 3 contain inconsistent data. Litter size in table 3 is not defined (it is not clear if the Authors mean total born, or live born). Anyway, the litter size in table 3 seem to be too low in comparison to the rest of data. In fact it is not clear what does it mean litter size, because lower row contain data from day 0 (so day of birth, what probably should be defined as litter size). However, the most important doubt is the lack of compatibility between table 2 and 3 (litter size, or d0 in table 3 contain data that has no equivalent in table 2 (nor total piglets born or piglets born alive). It is impossible to understand and must be clarified.
The Discussion is well written and Conclusion is generally consistent with data being presented.
To conclude, the manuscript describe interesting, and well prepared two phase experiment. The most of my comments are of minor importance, and probably could be easily amended. That is why I suggest minor revision.
Reviewer 2 Report
I agree that pig farmers would adopt any practices or products to improve the welfare of their pigs, if there is a demonstrated benefit to the pig and pursue the continual improvement of animal welfare.
As you say, results are not as expected due to minimal impacts from this product, and considering the cost to purchase and labour required to install.
I would like to indicate some comments and suggestions:
Line 229
These data do not agree with what is indicated in the table (8 grams heavier)
Line 237
For me, the explanation of Figure 3 is not clear. According to the text, A and B explain the behavior of piglets on day 2 of age. So, where were a greater number of piglets in the creep area or next to the udder compared to the same time?
Line 241
From my point of view it would make it easier to compare the graphs in Figure 3 if a title were written. For instance, A: piglets in the creep area on day 2 of age; B: piglets next to the udder on day 2 of age and so on, although it is explained at the bottom of the figure.
Line 251
The same nomenclature should be used: "Total litter size"(in the text) or "Total piglets born" in Table 2.
Line 253
Table 2 shows parity effects and nothing is explained. Why? At least a brief reference to the results.
Line 262
In Table 3: written abbreviation of average: avg
Line 273 to 275
See line 237
Reviewer 3 Report
An overall very interesting study which describes results of two trials performed in sows. A few minor corrections/additions are required.
